# Cross-modal Representation Flattening for Multi-modal Domain Generalization

**Yunfeng Fan**[1], **Wenchao Xu**[1,*], **Haohao Wang**[2], **Song Guo**[3]
[1]Department of Computing, The Hong Kong Polytechnic University,
[2]School of Computer Science and Technology, Huazhong University of Science and Technology,
[3]Hong Kong University of Science and Technology
yunfeng.fan@connnect.polyu.hk, wenchao.xu@polyu.edu.hk,
hz_wang@hust.edu.cn, songguo@cse.ust.hk

## Abstract

Multi-modal domain generalization (MMDG) requires that models trained on multi-modal source domains can generalize to unseen target distributions with the same modality set. Sharpness-aware minimization (SAM) is an effective technique for traditional uni-modal domain generalization (DG), however, with limited improvement in MMDG. In this paper, we identify that *modality competition* and *discrepant uni-modal flatness* are two main factors that restrict multi-modal generalization. To overcome these challenges, we propose to construct consistent flat loss regions and enhance knowledge exploitation for each modality via cross-modal knowledge transfer. Firstly, we turn to the optimization on representation-space loss landscapes instead of traditional parameter space, which allows us to build connections between modalities directly. Then, we introduce a novel method to flatten the high-loss region between minima from different modalities by interpolating mixed multi-modal representations. We implement this method by distilling and optimizing generalizable interpolated representations and assigning distinct weights for each modality considering their divergent generalization capabilities. Extensive experiments are performed on two benchmark datasets, EPIC-Kitchens and Human-Animal-Cartoon (HAC), with various modality combinations, demonstrating the effectiveness of our method under multi-source and single-source settings. Our code is open-sourced [1].

## 1 Introduction

Domain generalization (DG) aims to equip models with the ability to perform robustly across unseen domains when trained only on several source domains, thereby enhancing their adaptability and utility in real-world scenarios, such as autonomous driving [1, 2], medical health [3, 4], person re-identification [5, 6] and brain-computer interface [7, 8]. Methods on how to deal with domain shift have been extensively proposed in the literature, including domain alignment [9], meta-learning [10, 11], data augmentation [12, 13] and ensemble learning [14]. Despite the remarkable achievements of DG in recent years, most of research still focuses on uni-modal data. The emergence of various multi-modal datasets and the requirement to complete a variety of multi-modal tasks highlight the need to address multi-modal domain generalization (MMDG) problems.

Due to the complementary information that exists between modalities, MMDG aims to exploit generalization capabilities from each modality simultaneously. According to [15], the generalization capability of deep neural networks (DNNs) is closely related to their flatness of minima on loss

---

*Corresponding Author.

[1]https://github.com/fanyunfeng-bit/Cross-modal-Representation-Flattening-for-MMDG

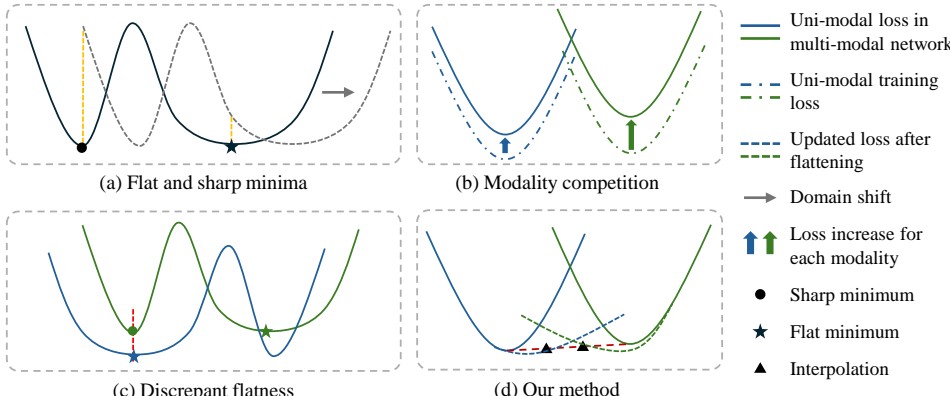

Figure 1: (a) Flat minima on loss landscape generalize better than sharp minima with domain shift. (b) Multi-modal joint training leads to larger loss for each modality compared with independent uni-modal training. (c) The flat minima between modalities are usually inconsistent, making it hard to obtain flat minima for each modality simultaneously in a multi-modal network. (d) We optimize the cross-modal interpolations on representation-space loss landscape to get consistent flat region.

landscape (as shown in Fig. 1 (a)), which motivates penalizing sharpness [16] and rewarding flatness [17]. Sharpness-aware minimization (SAM) [18] and its variants [14, 19] have been proposed to seek flatter minima and achieve better generalization across domains. Despite their success on uni-modal scenarios, in this paper, we argue that they are not compatible well in MMDG since the distinct properties between modalities pose two challenges (more details can be found in Sec.3.2). (1) **Modality competition**: according to [20], multiple modalities will compete with each other during joint training, leading to inadequate knowledge exploitation for each modality [21, 22], i.e, larger minima of loss as shown in Fig. 1 (b), and consequently worse generalization. (2) **Discrepant uni-modal flatness**: the generalization gap between modalities makes it hard to find their flat minima simultaneously, resulting in multi-modal networks incapable of utilizing generalization capabilities from all modalities, as illustrated in Fig. 1 (c). Hence, existing methods can not fully exploit the generalization potential of each modality, which inevitably leads to sub-optimal solutions for MMDG.

To overcome these challenges, we propose to construct consistent flat loss regions and enhance knowledge exploitation for each modality via cross-modal knowledge transfer. Traditional SAM-based methods are analyzed on parameter space. However, due to the heterogeneity between modalities, their parameter spaces could be extremely different (e.g., different model structures and parameter numbers), making it challenging to represent their correlation. Instead, we turn to optimization on representation-space loss landscape [23] as representations of different modalities can be mapped into a shared space, so that we can build their connections directly. Based on this, we propose a novel **C**ross-**M**odal **R**epresentation **F**lattening (CMRF) method to achieve consistent representation flat minima. As shown in Fig. 1 (d), we construct the interpolations by mixing paired multi-modal representations and then optimize them to flatten the high-loss regions between minima from different modalities. Specifically, we obtain more stable and generalizable cross-modal interpolations from moving averaged teacher model and then employ feature distillation to regularize the learning of each modality. The interpolations between modalities bring their flat regions closer, alleviating their flatness discrepancy. Moreover, the cross-modal knowledge transfer also helps to promote each modality and alleviate their competition. Our contributions can be summarized as:

- To the best of our knowledge, we are the first to extend the uni-modal flatness analysis to MMDG, and empirically attribute the reasons for limited MMDG performance to two problems: modality competition and discrepant uni-modal flatness.

- We construct shared representation space instead of parameter space to build connections between modalities directly and propose to flatten high-loss representation regions between modalities by interpolating mixed multi-modal representations and performing knowledge distillation to regularize the learning of each modality.

- Extensive experiments verify the effectiveness and superiority of our framework on two benchmark datasets of EPIC-Kitchens and Human-Animal-Cartoon (HAC) under various modalities combinations on both multi- and single-source MMDG.

## 2   Related Work

**Flat Minimum of Loss Landscape for DG.** Domain generalization refers to the ability of models to perform well on new, unseen domains that are dissimilar with domains they were trained on. Numerous methods have been proposed to tackle the domain shift, while one type among them is to search for flat minima in loss landscapes [18, 24, 19]. Jiang *et al.* [15] conducted comprehensive measures and found that a sharpness-based measure has highest correlation with generalization. Based on that, Foret *et al.* [18] proposed sharpness-aware minimization (SAM) to seek parameters that lie in neighborhoods with uniformly low loss via perturbed gradients, while Wang *et al.* [25] further proposed to align the gradient directions between the empirical risk and the perturbed loss. Moreover, average weights during training has also shown to yield flatter minima [17], which motivates more elegant average methods such as SWAD [14] and EoA [26]. In this paper, we try to optimize consistent flat minima for different modalities in representation-space loss landscapes instead of traditional parameter space.

**Multi-modal DG.** Although uni-modal DG has been extensively studied in recent years, the research on MMDG is severely insufficient, while only few works have been done. Planamente *et al.* [27] proposed RNA-Net to balance audio and video feature norms via a relative norm alignment loss. Dong *et al.* [28] proposed a unified framework to achieve domain generalization in various multi-modal scenarios including multi-source, uni-source, and modality missing DG. In this paper, we extend the uni-modal flatness analysis to MMDG and address two particular problems in multi-modal scenarios.

**Mixup.** Mixup [29] is a data augmentation technique introduced to improve the generalization performance of models. Traditional mixup and its variant CutMix [30] are performed on input data, while Verma *et al.* [31] further introduced Manifold Mixup that mixes the representations in each layer to produce smoother decision boundaries. However, Manifold Mixup and its variants [32, 33] are designed for uni-modal data, and only few works are on multi-modal scenarios [34, 35]. STEMM [34] aims to align speech and text features by mixing them, but is limited with its architecture-specific design. Oh *et al.* [35] introduced $m^2$-Mix aiming at generating hard negative samples by mixing image and text embeddings to fine-tuning CLIP. Compared with them, our mixed multi-modal representations has no architecture restrictions and are used as teacher signals to guide various modalities to learn consistent flat minima.

## 3   Method

### 3.1   Preliminaries

We follow the definition of multi-modal domain generalization problem as in [28]. In MMDG, we are given $D$ source domains for training $\mathcal{D}_{train} = \left\{ \mathcal{D}^i | i = 1, \cdots, D \right\}$, where $\mathcal{D}^i = \left\{ \left( \mathbf{x}_j^i, y_j^i \right) \right\}_{j=1}^{n_i} \sim P_{XY}^i$ denotes the $i$-th domain with $n_i$ data instances sampled from a joint distribution of input samples and output labels $P_{XY}^i$. $X$ and $Y$ represent the corresponding random variables. Each input instance $\mathbf{x}_j^i = \left\{ \left( \mathbf{x}_j^i \right)_k | k = 1, \cdots, M \right\} \in \mathbf{X}$ consists of $M$ different modalities and $y_j^i \in \mathcal{Y} \subset \mathbb{R}$ denotes corresponding label, where $\mathbf{X}$ and $\mathcal{Y}$ represent input and output space. The joint distributions in $\mathcal{D}_{train}$ are different from each other: $P_{XY}^i \neq P_{XY}^j, 1 \leq i \neq j \leq D$. Now, with an unseen test domain $\mathcal{D}_{test}$ with $M$ modalities that cannot be accessed during training and $P_{XY}^{test} \neq P_{XY}^i$ for $i \in \{1, \cdots, D\}$, the goal of MMDG is to learn a robust and generalizable predictive function $f : \mathbf{X} \to \mathcal{Y}$ based on $D$ training domains to achieve a minimum prediction error on $\mathcal{D}_{test}$:

$$\min_f \mathbb{E}_{(\mathbf{x},y) \in \mathcal{D}_{test}} \left[ \ell \left( f \left( \mathbf{x} \right), y \right) \right] \tag{1}$$

where $\mathbb{E}$ is the expectation and $\ell \left( \cdot, \cdot \right)$ is the loss function, e.g., cross-entropy loss for multi-modal classification tasks. In this paper, we use $\theta = \{\theta_1, \cdots, \theta_M\}$ to denote the parameters of the neural network $f$, where $\theta_i$ indicates the parameters for $i$-th modality. Therefore, the training loss over all training domains $\mathcal{D}_{train}$ is defined as follows:

$$\mathcal{L} \left( \theta; \mathcal{D}_{train} \right) = \frac{1}{\sum_{i=1}^D n_i} \sum_{i=1}^D \sum_{j=1}^{n_i} \ell \left( f \left( \mathbf{x}_j^i; \theta \right), y_j^i \right) \tag{2}$$

Table 1: MMDG analysis on EPIC-Kitchens and HAC with video and audio data. 'Base' denotes the naive multi-modal joint training without any domain generalization strategies. 'Uni-video' and 'Uni-audio' means training only with uni-modal data. 'Video', 'Audio' and 'Video-Audio' denote testing with uni-modal and multi-modal data. Results are averaged by using each domain as target.

| | EPIC-Kitchens | | | HAC | | |
|---|---|---|---|---|---|---|
| | Video | Audio | Video-Audio | Video | Audio | Video-Audio |
| Uni-video | 58.73 | - | - | 68.07 | - | - |
| Uni-audio | - | 40.04 | - | - | 32.81 | - |
| Uni-video-SAM | **61.68** | - | - | 69.58 | - | - |
| Uni-audio-SAM | - | 42.65 | - | - | **35.84** | - |
| Base | 56.65 | 38.62 | 59.63 | 67.60 | 31.24 | 63.11 |
| SAM | 58.80 | 37.77 | 61.19 | 68.46 | 31.56 | 64.72 |
| CMRF (ours) | 60.66 | **43.13** | **63.91** | **70.54** | 34.86 | **71.91** |

The empirical risk minimization (ERM) of Eq. 2 tends to converge to sharp minima and SAM [18] is proposed to seek flatter minima on loss landscape with the following optimization:

$$\min_{\theta} \mathcal{L}\left(\theta + \hat{\epsilon}; \mathcal{D}_{train}\right), \text{ where } \hat{\epsilon} \triangleq \rho \frac{\nabla \mathcal{L}\left(\theta; \mathcal{D}_{train}\right)}{\left\|\mathcal{L}\left(\theta; \mathcal{D}_{train}\right)\right\|}. \tag{3}$$

where $\rho$ is a predefined constant controlling the radius of the neighborhood.

### 3.2 MMDG Analysis

MMDG aims to comprehensively exploit the generalization capabilities from each modality to learn more robust and generalized models. However, the generalization behavior of each modality in multi-modal networks has not been well explored. Here, we analyze the behavior of each modality and find the challenges for generalizable multi-modal networks.

**Modality competition leads to larger minima.** As demonstrated in Tab. 1, we compare naive joint training and SAM about their uni- and multi-modal performance. SAM can clearly improve generalization on both uni-modal and multi-modal training. However, the uni-modal generalization from multi-modal trained network is worse than uni-modal trained network, whether or not SAM is applied (e.g, 56.65% vs. 58.73% without SAM and 58.80% vs. 61.68% with SAM on EPIC-Kitchens video). This phenomenon can be explained by modality competition [20, 36] that modalities in joint training compete with each other, making each modality under-explored. Our empirical results show that it not only degrades in-domain performance for each modality as discussed in [37, 38], but also weakens their out-of-domain generalization, resulting in larger minima of loss as shown in Fig. 1 (b).

**Generalization gap results in discrepant uni-modal flatness.** We observe that applying SAM can only improve generalization of better modality in multi-modal network but has marginal benefit or even harm on weak modality (e.g., video generalization is improved from 56.65% to 58.80% on EPIC-Kitchens while the number of audio drops from 38.62% to 37.77%). According to [38], the better modality will dominate multi-modal gradients. Hence, in Eq. 3, the gradient perturbation $\hat{\epsilon}$ in SAM could also be dominated by the better modality, which means this optimization on multi-modal network tends to search for flatter regions for modality with better generalization but ignores other weak modalities. This suggests that conventional uni-modal SAM-based methods cannot find the coexisting flat minima for each modality due to their generalization gap, leading to discrepant flatness and consequently under-utilization of generalization from all modalities, as shown in Fig. 1 (c). More results with other modality combinations can be found in Sec. 4.2 and Appendix. B.

### 3.3 Cross-Modal Representation Flattening

Based on the analyses above, in this paper, we aim to 1) accomplish consistent flat minima for all modalities in multi-modal network and 2) alleviate the competition between modalities to utilize their generalization comprehensively. Considering the correlation and complementary information between modalities, we propose to leverage cross-modal knowledge transfer to enhance MMDG.

**Representation-space loss landscape.** Previous analysis of loss landscapes usually happens on parameter space [19, 39]. However, the network structures and sizes for different modalities are

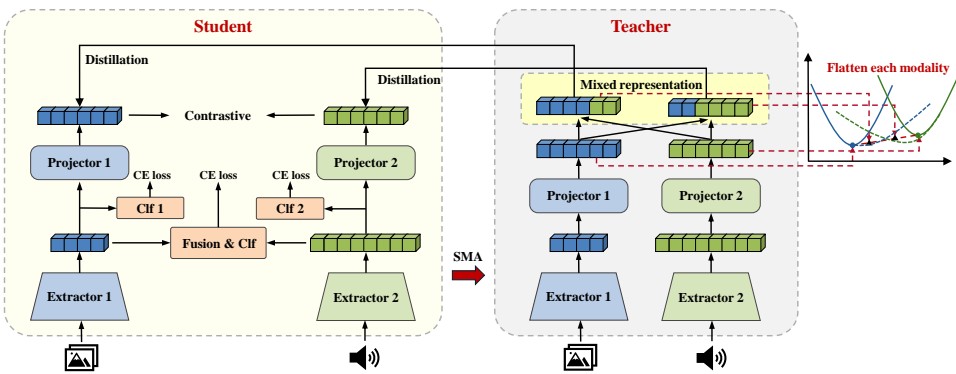

Figure 2: The overall framework of our method. The projectors map features with different dimensions to the same representation space. The teacher model is moving averaged from online model and generates cross-modal mixed representations as interpolations to distill the student representations. Uni-modal classifier is used to lower the loss of distilled features for each modality and a contrastive loss aims to alleviate gap between modalities. Only the online student model back propagates gradients. **The teacher model is used for evaluation finally.**

commonly different, leading to disparate parameter spaces. This makes it difficult to catch correlations between modalities and produce consistent flat loss regions in parameter space. Inspired by [23] that introduces representation-space loss landscape, we turn to analyze loss landscapes of different modalities in representation space. Specifically, given a data point $\mathbf{x}_j^i = \left\{ \left( \mathbf{x}_j^i \right)_k | k = 1, \cdots, M \right\}$, feature extractors are usually applied to transform input data into features with different dimensions:

$$\left( \boldsymbol{h}_j^i \right)_k = g_k \left( \left( \mathbf{x}_j^i \right)_k \right) \subset \mathbb{R}^{d_k} \tag{4}$$

where $g_k$ is feature extractor for $k$-th modality, $d_k$ is feature dimension size and $\exists k \neq l, d_k \neq d_l$. In this paper, we use a projector $Proj_k \left( \cdot \right)$ for $k$-th modality that maps its features into a shared representation space for all modalities with the same dimension $d$ (omit superscript and subscript of domain and instance index for simplicity):

$$\boldsymbol{z}_k = Proj_k \left( \boldsymbol{h}_k \right) \subset \mathbb{R}^d, \ k \in \{1, \cdots, M\} \tag{5}$$

Given that each point in the representation space corresponds to a specific loss value, it is feasible to construct a landscape that maps each representation point to its associated loss value (e.g., horizontal axis indicates representation and vertical axis indicates loss in Fig. 1 (d)). After training, each representation extracted from each training sample can be viewed as a minimum. And we can judge whether a representation minimum is flat or sharp according to its neighboring loss distribution. In the shared representation loss landscape, we can build connections between different modalities directly.

**Cross-modal representation interpolation.** As discussed in Sec. 3.2, the discrepant uni-modal flatness severely impedes the utilization of generalization capability from each modality. The conclusion also applies to representation-space loss landscape since better modality still dominates gradients of representations, which optimizes weak modalities at sharp regions. Therefore, to obtain flat minima for various modalities simultaneously, we aim to flatten the high-loss regions between minima from different modalities. Given the paired multi-modal representations $\boldsymbol{z}_k$ and $\boldsymbol{z}_l$, $k \neq l$, we construct interpolated representations between them by cross-modal representation mixup:

$$\boldsymbol{z}_{k,l} = \delta \boldsymbol{z}_k + (1 - \delta) \boldsymbol{z}_l \tag{6}$$

where $\delta$ is mixing ratio. If the loss of mixed representations can be optimized to lower values, we would get a flatter region between modalities, as demonstrated in Fig. 1 (d). However, according to [31], directly optimization on mixed representations requires mixup at multiple eligible layers to be effective. It is impractical in multi-modal scenarios because representations of each layer for different modalities are generally at different scales, converting all them into a shared space is costly. In this paper, we propose a simple yet effective method that distills the knowledge from mixed representations to each modality and then optimize the learned representations. Firstly, we perform

simple moving average (SMA) [26] for the online updated network $\theta_k$ of each modality to establish the teacher network $\hat{\theta}_k^t$, which can produce more stable and generalizable representations:

$$\hat{\theta}_k^t = \begin{cases} \theta_k^t, & \text{if } t \leq t_0 \\ \frac{t-t_0}{t-t_0+1} \cdot \hat{\theta}_k^{t-1} + \frac{1}{t-t_0+1}\theta_k^t, & \text{otherwise} \end{cases} \tag{7}$$

where $\theta_k^t$ is the online model's state at iteration $t$ of $k$-th modality. $t_0$ is the start iteration for SMA. Hence, the representation from teacher network is denoted as $\hat{z}_k$ and the mixed representation of Eq. 6 should be rewritten as:

$$\hat{z}_{k,l} = \delta\hat{z}_k + (1-\delta)\hat{z}_l, \ \delta \sim Beta(\alpha, \alpha) \tag{8}$$

where $\alpha$ is a hyperparameter in Beta distribution. Considering the semantic gap between modalities, we let **interpolation closer** to $k$-th modality act as its teacher signal, so distillation loss should be:

$$\begin{cases} \mathcal{L}_{dis}^k = \frac{1}{M-1}\sum_{l=1,l\neq k}^{M} \|z_k - \hat{z}_{k,l}\|_2^2, & \delta > 0.5 \\ \mathcal{L}_{dis}^l = \frac{1}{M-1}\sum_{k=1,k\neq l}^{M} \|z_l - \hat{z}_{k,l}\|_2^2, & \delta < 0.5 \end{cases} \tag{9}$$

Then, we assign specific classifier for each modality before $Proj_k(\cdot)$ to online models and optimize the features by classification loss $\mathcal{L}_{cls}^k$. **The combination $\mathcal{L}_{dis}^k + \mathcal{L}_{cls}^k$ flattens the neighboring representation-space loss landscape of $k$-th modality to other modalities.** Further, we employ a multi-modal supervised contrastive loss on shared representation space, which can help to narrow the gap between modalities and make it conducive to flatten the region between them. For a random batch $\mathcal{B}$ with $M \times B$ uni-modal samples, we let $i$ as the index of a uni-modal instance in the batch, and define $P(i)$ as the set of uni-modal samples that have the same label with $i$ (except itself). The supervised contrastive loss can be written as (notably, subscript here does not denote modality index but the index of each sample):

$$\mathcal{L}_{con} = \sum_{i\in\mathcal{B}} -\frac{1}{|P(i)|} \sum_{p\in P(i)} \log \frac{\exp(z_i \cdot z_p/\tau)}{\sum_{a\in\mathcal{B}\setminus\{i\}} \exp(z_i \cdot z_a/\tau)} \tag{10}$$

where $\tau \in \mathcal{R}^+$ is the temperature parameter.

**Adaptive weight.** As demonstrated in Tab. 1, the generalization capabilities between modalities may have significant gaps, so we propose to assign stronger flattening weights to better modalities. We compare the uni-modal validation accuracy from teacher model (calculated by the moving averaged uni-modal classifier) as a rough estimate of the difference in generalization ability between modalities (the performance of different modalities on in-domain validation set can generally reflect their strength in generalization capability, as shown in Appendix. B). The distillation loss can be modified as:

$$\mathcal{L}_{dis}^k = \frac{1}{M-1} \sum_{l=1,l\neq k}^{M} \eta_{k,l} \|z_k - \hat{z}_{k,l}\|_2^2, \ \eta_{k,l} = \begin{cases} 1 & \hat{A}_k/\hat{A}_l > \mu \\ 0.5 & \hat{A}_k/\hat{A}_l \leq \mu \end{cases} \tag{11}$$

where $\hat{A}_k$ denotes the validation accuracy of $k$-th modality by teacher model, $\mu$ is a hyperparameter (default 1.2 in this paper). In this way, the teacher signal with stronger generalization ability is applied with a larger distillation weight. Finally, we can get our final loss as follows:

$$\mathcal{L} = \mathcal{L}_{cls} + \sum_{k=1}^{M} \lambda_1 \mathcal{L}_{cls}^k + \sum_{k=1}^{M} \lambda_2 \mathcal{L}_{dis}^k + \lambda_3 \mathcal{L}_{con} \tag{12}$$

where $\mathcal{L}_{cls}$ is the multi-modal classification loss, and $\lambda_1$, $\lambda_2$ and $\lambda_3$ are hyperparameters to control the strength of each loss. Finally, we use teacher model for evaluation as it averages learned knowledge from student for better generalization.

## 4 Experiments

### 4.1 Experimental Setting

**Dataset and implementation details.** We utilize two benchmark datasets, EPIC-Kitchens [40] and Human-Animal-Cartoon (HAC) [28], both of them have video, optical flow, and audio data. Three

Table 2: Multi-modal **multi-source** DG with different modalities on EPIC-Kitchens and HAC datasets. The best is in **bold**, and the second best is underlined.

| | Modality | | | EPIC-Kitchens | | | | HAC | | | |
|---|---|---|---|---|---|---|---|---|---|---|---|
| Method | Video | Audio | Flow | D2, D3 → D1 | D1, D3 → D2 | D1, D2 → D3 | Avg | A, C → H | H, C → A | H, A → C | Avg |
| Base | ✓ | ✓ | | 54.94 | 62.26 | 61.70 | 59.63 | 69.92 | 69.32 | 50.09 | 63.11 |
| SAM [18] | ✓ | ✓ | | 55.86 | 63.33 | 64.37 | 61.19 | 64.49 | 76.70 | 52.96 | 64.72 |
| SAGM [25] | ✓ | ✓ | | 56.81 | 65.10 | 65.33 | 62.08 | 71.17 | 72.05 | 55.38 | 66.20 |
| SWAD [14] | ✓ | ✓ | | 55.63 | 63.74 | 63.55 | 60.97 | 70.72 | 72.94 | 53.45 | 65.70 |
| EoA [26] | ✓ | ✓ | | 55.63 | 64.93 | 64.68 | 61.75 | 69.20 | 77.27 | 58.71 | 68.39 |
| RNA-Net [27] | ✓ | ✓ | | 55.37 | 64.20 | 62.25 | 60.61 | 67.45 | 68.32 | 54.78 | 63.52 |
| SimMMDG [28] | ✓ | ✓ | | 57.24 | 65.07 | 63.55 | 61.95 | 72.75 | 76.14 | 54.59 | 67.83 |
| CMRF (ours) | ✓ | ✓ | | 56.55 | 68.13 | 67.04 | 63.91 | 76.45 | 82.39 | 56.88 | 71.91 |
| Base | ✓ | | ✓ | 55.86 | 67.47 | 59.34 | 60.89 | 72.83 | 77.84 | 43.58 | 64.75 |
| SAM [18] | ✓ | | ✓ | 58.85 | 67.33 | 63.96 | 63.38 | 74.27 | 78.98 | 46.79 | 66.68 |
| SAGM [25] | ✓ | | ✓ | 57.64 | 66.70 | 64.67 | 63.00 | 76.78 | 75.10 | 45.80 | 65.89 |
| SWAD [14] | ✓ | | ✓ | 59.79 | 67.33 | 62.47 | 63.20 | 75.82 | 78.33 | 51.90 | 68.68 |
| EoA [26] | ✓ | | ✓ | 62.99 | 68.89 | 63.76 | 65.21 | 74.45 | 80.68 | 53.13 | 69.42 |
| RNA-Net [27] | ✓ | | ✓ | 54.21 | 64.80 | 59.31 | 59.44 | 74.56 | 75.39 | 44.90 | 64.95 |
| SimMMDG [28] | ✓ | | ✓ | 57.03 | 66.67 | 63.86 | 62.82 | 77.90 | 78.98 | 57.80 | 71.56 |
| CMRF (ours) | ✓ | | ✓ | 65.28 | 67.87 | 64.89 | 66.01 | 81.16 | 81.25 | 55.50 | 72.64 |
| Base | | ✓ | ✓ | 49.42 | 55.60 | 54.41 | 53.14 | 52.89 | 55.11 | 40.92 | 49.64 |
| SAM [18] | | ✓ | ✓ | 54.48 | 59.87 | 57.90 | 57.42 | 54.71 | 59.66 | 47.21 | 53.86 |
| SAGM [25] | | ✓ | ✓ | 55.76 | 61.32 | 60.28 | 59.11 | 55.90 | 61.03 | 47.48 | 54.80 |
| SWAD [14] | | ✓ | ✓ | 51.32 | 61.74 | 61.05 | 58.04 | 54.71 | 59.76 | 52.00 | 55.49 |
| EoA [26] | | ✓ | ✓ | 52.41 | 60.67 | 61.81 | 58.30 | 55.43 | 58.97 | 52.29 | 55.56 |
| RNA-Net [27] | | ✓ | ✓ | 50.89 | 54.24 | 55.90 | 53.68 | 53.11 | 59.32 | 43.82 | 52.08 |
| SimMMDG [28] | | ✓ | ✓ | 55.86 | 64.60 | 59.34 | 59.93 | 57.88 | 60.79 | 48.62 | 55.76 |
| CMRF (ours) | | ✓ | ✓ | 57.24 | 64.94 | 66.12 | 62.76 | 59.06 | 61.79 | 55.04 | 58.49 |
| Base | ✓ | ✓ | ✓ | 54.71 | 67.20 | 61.70 | 61.20 | 70.29 | 71.25 | 53.57 | 65.07 |
| SAM [18] | ✓ | ✓ | ✓ | 56.78 | 65.20 | 62.22 | 61.40 | 75.36 | 73.68 | 57.34 | 68.79 |
| SAGM [25] | ✓ | ✓ | ✓ | 57.76 | 67.12 | 61.78 | 62.22 | 76.56 | 75.48 | 56.92 | 69.65 |
| SWAD [14] | ✓ | ✓ | ✓ | 55.84 | 68.21 | 64.90 | 62.98 | 75.78 | 74.95 | 58.02 | 69.58 |
| EoA [26] | ✓ | ✓ | ✓ | 57.93 | 68.53 | 68.78 | 65.08 | 76.09 | 76.95 | 57.19 | 70.08 |
| RNA-Net [27] | ✓ | ✓ | ✓ | 56.25 | 63.47 | 59.72 | 59.81 | 71.89 | 70.88 | 54.58 | 65.78 |
| SimMMDG [28] | ✓ | ✓ | ✓ | 62.08 | 66.13 | 64.40 | 64.20 | 76.27 | 77.70 | 56.42 | 70.13 |
| CMRF (ours) | ✓ | ✓ | ✓ | 61.84 | 70.13 | 70.12 | 67.36 | 78.26 | 79.54 | 60.09 | 72.44 |

Table 3: Multi-modal **single-source** DG with video, flow and audio three modalities on EPIC-Kitchens and HAC datasets.

| | | EPIC-Kitchens | | | | | | | HAC | | | | | | |
|---|---|---|---|---|---|---|---|---|---|---|---|---|---|---|---|
| | Source: | D1 | | D2 | | D3 | | | H | | A | | C | | |
| Method | Target: | D2 | D3 | D1 | D3 | D1 | D2 | Avg | A | C | H | C | H | A | Avg |
| Base | | 56.80 | 53.08 | 47.36 | 59.65 | 55.63 | 56.93 | 54.91 | 64.20 | 39.45 | 64.85 | 52.29 | 57.97 | 65.90 | 57.44 |
| SAM [18] | | 54.40 | 55.24 | 49.65 | 61.40 | 54.94 | 65.33 | 56.83 | 67.61 | 44.04 | 66.67 | 60.09 | 60.14 | 61.36 | 59.98 |
| SAGM [25] | | 53.11 | 57.32 | 50.46 | 60.12 | 56.79 | 65.10 | 57.15 | 67.86 | 45.31 | 64.90 | 57.35 | 64.10 | 63.16 | 60.45 |
| SWAD [14] | | 57.46 | 56.92 | 50.46 | 63.33 | 56.25 | 64.58 | 58.17 | 68.43 | 43.79 | 68.32 | 57.35 | 62.80 | 67.37 | 61.34 |
| EoA [26] | | 58.40 | 57.39 | 51.26 | 64.58 | 55.17 | 63.33 | 58.35 | 68.18 | 44.95 | 69.94 | 56.88 | 67.39 | 69.02 | 62.73 |
| RNA-Net [27] | | 50.32 | 51.27 | 48.90 | 61.34 | 53.76 | 55.89 | 53.58 | 62.35 | 43.24 | 64.21 | 53.46 | 55.37 | 66.82 | 57.57 |
| SimMMDG [28] | | 54.13 | 57.90 | 50.57 | 63.04 | 60.69 | 64.27 | 58.43 | 64.77 | 39.44 | 71.38 | 50.46 | 60.14 | 70.77 | 59.49 |
| CMRF (ours) | | 60.80 | 56.78 | 55.17 | 64.99 | 57.24 | 65.73 | 60.12 | 68.75 | 46.33 | 73.55 | 58.26 | 65.22 | 72.46 | 64.09 |

distinct domains for EPIC-Kitchens are D1, D2, and D3 and for HAC are humans (H), animals (A), and cartoon figures (C). Our experiment setup follow [28]. Training details including model structures, hyperparameters, and experimental environment can be found in Appendix. A.

**Baselines.** We compare our CMRF with seven different baselines that can be divided into four groups: 1) Base, naive multi-modal joint training without any domain generalization strategies, 2) SAM [18] and SAGM [25], searching for flat minima in parameter loss landscapes, 3) SWAD [14] and EoA [26], ensemble-based methods for flat minima, and 4) RNA-Net [27] and SimMMDG [28], domain generalization methods specifically designed for MMDG. SAM, SAGM, SWAD and EoA are initially designed for uni-modal DG and we extent them into MMDG. For all methods, we follow [41] and select the model with best validation (in-domain) accuracy to evaluate generalization on test (out-of-domain) data. We report the Top-1 accuracy for all results.

Table 4: The average results of uni-modal performance comparison under multi-modal multi-source DG on EPIC-Kitchens with different modality combinations.

| | Video | Audio | Video-Audio | Video | Flow | Video-Flow | Flow | Audio | Flow-Audio |
|---|---|---|---|---|---|---|---|---|---|
| Uni-video | 58.73 | - | - | 58.73 | - | - | - | - | - |
| Uni-flow | - | - | - | - | 58.30 | - | 58.30 | - | - |
| Uni-audio | - | 40.04 | - | - | - | - | - | 40.04 | - |
| Base | 56.65 | 38.62 | 59.63 | 55.28 | 55.78 | 60.89 | 54.86 | 39.42 | 53.14 |
| SAM [18] | 58.80 | 37.77 | 61.19 | 59.76 | 56.05 | 64.05 | 56.82 | 40.35 | 57.42 |
| EoA [26] | 57.54 | 39.70 | 61.75 | 57.49 | 57.17 | 65.21 | 57.32 | 40.14 | 58.30 |
| SimMMDG [28] | 59.43 | 38.43 | 61.95 | 57.02 | 55.60 | 62.82 | 58.21 | 40.03 | 59.93 |
| CMRF (ours) | **60.66** | **43.13** | **63.91** | **59.83** | **58.33** | **66.01** | **59.63** | **43.58** | **62.76** |

Table 5: Ablations of each module on EPIC-Kitchens with video and audio data. DL: distillation loss, UCL: uni-modal classification loss, CL: contrastive loss, AW: adaptive weight, SMA: simple moving average.

| DL | UCL | CL | AW | SMA | D2, D3 → D1 | D1, D3 → D2 | D1, D2 → D3 | Avg |
|---|---|---|---|---|---|---|---|---|
| | | | | | 54.94 | 62.26 | 61.70 | 59.63 |
| ✓ | | | | | 55.63 | 63.87 | 62.14 | 60.55 |
| | ✓ | | | | 53.10 | 64.12 | 64.70 | 60.64 |
| ✓ | ✓ | | | | 52.75 | 66.33 | 65.21 | 61.43 |
| ✓ | ✓ | ✓ | | | 55.79 | 65.65 | 63.92 | 61.79 |
| ✓ | ✓ | ✓ | ✓ | | 53.84 | 66.79 | 66.14 | 62.26 |
| ✓ | ✓ | ✓ | | ✓ | 55.79 | 67.53 | 65.21 | 62.84 |
| ✓ | ✓ | ✓ | ✓ | ✓ | **56.55** | **68.13** | **67.04** | **63.91** |

Table 6: Ablation studies on interpolated representations on HAC with video and audio data. SM dis: self-modal distillation, CM dis: cross-modal distillation, Fixed Mix: interpolations with fixed mixing ratio (0.5-0.5).

| Method | D2, D3 → D1 | D1, D3 → D2 | D1, D2 → D3 | Avg |
|---|---|---|---|---|
| SM dis | 74.37 | 80.68 | 56.42 | 70.49 |
| CM dis | 75.72 | 78.85 | 54.13 | 69.57 |
| Fixed Mix | 75.26 | 81.81 | 53.21 | 70.09 |
| Rand Mix (ours) | **76.45** | **82.39** | **56.88** | **71.91** |

## 4.2 Main Results

**Multi-modal multi-source DG.** Tab. 2 illustrate the results of our CMRF and all baselines on EPIC-Kitchens and HAC under multi-modal multi-source domain generalization setting, where the models are trained on multiple source domains and test on one target domain. We conduct experiments by combining any two modalities, as well as all three modalities, to validate the generalization of our method. As we can see from Tab. 2, our CMRF outperforms all baselines on almost all settings and achieves great improvement on the average results (by up to 3.52% with video-audio modalities on HAC). The uni-modal DG methods, especially SAGM and EoA, can improve the generalization of multi-modal network to a certain extent, but their improvements are limited as they do not consider modality competition and inconsistent flatness between modalities. Two MMDG methods RNA-Net and SimMMDG also perform less than satisfactory since they do not fully exploit the generalization capability of each modality.

**Multi-modal single-source DG.** Our CMRF does not requires domain labels for training, making it feasible to perform multi-modal single-source domain generalization, where models are trained on a single source domain and test on other multiple target domains. The results trained with three modalities are presented in Tab. 3. Our CMRF still apparently outperforms all baselines on average accuracy, despite being trained only on single-source domain data. For baselines with domain generalization strategies, they can not improve consistently across datasets, e.g., SimMMDG achieves the second best on EPIC-Kitchens but has limited improvement on HAC, showing their unstable generalization and their limitations in the single-source DG setting.

**Uni-modal performance in MMDG.** As we discussed in Sec. 3.2, exploiting the generalization capability of each modality simultaneously is the key to improving multi-modal domain generalization performance. Therefore, we evaluate the uni-modal performance from multi-modal trained networks to show the superiority of our method. We freeze the trained uni-modal feature extractor and train a linear classifier to test uni-modal performance. The results of average multi-source accuracy on EPIC-Kitchens are shown in Tab. 4. We can see that our CMRF not only improves the multi-modal domain generalization, but also greatly promotes its uni-modal domain generalization, even better than that of uni-modal training (60.66% vs. 58.73% and 43.12% vs. 40.04% for video and audio on EPIC-Kitchens), indicating the effectiveness of CMRF to use cross-modal knowledge to promote the

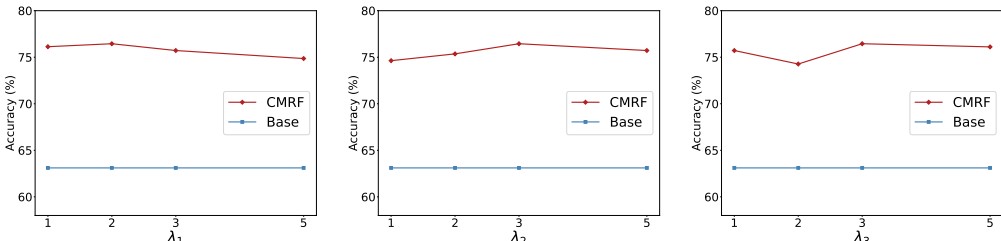

Figure 3: Parameter sensitivity analysis on HAC with video and audio data under A, C → H.

generalization of each modality via mitigating modality competition and flattening representation loss landscape between modalities. In Appendix B, we show the alleviated competition under in-domain performance and flatter region with perturbations. As for baselines, SAM and SimMMDG only enhance the generalization of better modality and EoA just achieves marginal uni-modal improvement, which means they can not utilize the generalization capability of all modalities comprehensively. Detailed results for each test domain and more results on HAC dataset are shown in Appendix. B.

### 4.3 Ablation Studies

**Ablation on each design.** Our CMRF contains five main modules: distillation loss $\mathcal{L}_{dis}^k$, uni-modal classification loss $\mathcal{L}_{cls}^k$, multi-modal supervised contrastive loss $\mathcal{L}_{con}$, adaptive weight, and SMA for teacher model. We conduct extensive ablation experiments to verify the effectiveness of each proposed module on EPIC-Kitchens with video-audio data under multi-source domain generalization setting. The results are illustrated in Tab. 5. Only applying distillation loss or uni-modal classification loss improves slightly and their combination leads to noticeable increase, highlighting the importance of flattening representation loss landscape between modalities for domain generalization. However, it does not guarantee steady improvement, e.g., the accuracy decreases from 54.94% to 52.75% in D2, D3 → D1 setting. Multi-modal supervised contrastive loss can enhance the average generalization by a small margin. Adaptive weight and using SMA network as teacher can both improve MMDG by a large margin, suggesting that it is necessary to emphasize the more generalized modality and obtain more stable distillation signals. Finally, combining all of them achieves the best results for multi-modal domain generalization, hence, each of them is indispensable.

**Ablation on interpolations.** In this paper, we mix multi-modal representations in the random ratio generated from Beta distribution as teacher signals, and choose interpolations closer to current modality for distillation, as in Eq. 9. We conduct experiments by using different forms of teacher signals to verify our method's effectiveness, as presented in Tab. 6. For $k$-th modality, we set $\delta$ to 1, 0, 0.5 for self-modal distillation, cross-modal distillation, and distillation with fixed mixing ratio. Since self-modal distillation can enhance learning for each modality via more generalizable signals, it achieves great performance next to ours. The heterogeneous knowledge between modalities makes cross-mode distillation worse. Fixed mixing ratio only locates one interpolation while our random ratio covers all possible points, resulting in our better performance.

Table 7: The average results compared with methods designed for modality competition on HAC with video and audio data under multi-source DG.

|  | Validation | Test |
|---|---|---|
| Base | 91.41 | 63.11 |
| Grad Blending [42] | 92.70 | 66.82 |
| OGM-GE [37] | 93.67 | 64.33 |
| PMR [38] | **94.90** | 65.24 |
| CMRF | 93.21 | **71.91** |

**Comparison with methods designed for modality competition.** Here, we conduct experiments with three baselines Gradient Blending [42], OGM-GE [37], and PMR [38] for modality competition as we attribute it as one challenge for MMDG. We not only report out-of-domain test accuracy but also in-domain validation results, as shown in Tab. 7. We can see that these methods can actually promote their performance on multi-modal validation set since they mitigate the competition. However, they tend to locate at sharp minima and the generalization gap between modalities still makes it hard to build consistent flat minima for different modalities. Hence, their performance increase on test set is limited, while our method achieves significant improvement on both validation and test sets.

**Parameter sensitivity.** Fig. 3 shows the results of different values on loss weights $\lambda_1$, $\lambda_2$, and $\lambda_3$. Since our method uses the moving averaged teacher model for evaluation, it is insensitive to hyperparameters.

## 5    Conclusion

In this paper, we analyze the behavior of multi-modal domain generalization and find that modality competition and discrepant uni-modal flatness restrict the generalization capability of multi-modal network. To address these challenges, we propose cross-modal representation flattening (CMRF) to construct consistent flat regions in a shared representation-space loss landscape. Our method builds interpolations by mixing multi-modal representations from moving averaged teacher model and use feature distillation to optimize the high-loss regions between modalities. Our extensive experiments on two benchmark datasets demonstrate the effectiveness of our method to promote multi-modal domain generalization, as well as uni-modal domain generalization in multi-modal network.

**Limitations.** Currently, we need to test on validation set to estimate generalization of each modality for Eq. 11, which can be time-consuming with the scale increase of validation set. In future work, we can add low-frequency noise as in [23] for domain shifting to evaluate the generalization.

## Acknowledgments and Disclosure of Funding

This work described in this paper is supported by two grant from the Research Grants Council of the Hong Kong Special Administrative Region, China (Project No. PolyU15222621, PolyU15225023) and National Natural Science Foundation of China under grants 62302184.

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

# A    Experimental Setting

**Dataset.** We utilize two benchmark datasets: EPIC-Kitchens [40] and Human-Animal-Cartoon (HAC) [28]. Our experimental setup follows the protocols established for the EPIC-Kitchens dataset in [43] and for the HAC dataset in [28]. The EPIC-Kitchens dataset encompasses eight actions ('put', 'take', 'open', 'close', 'wash', 'cut', 'mix', and 'pour') captured across three different kitchens, forming three distinct domains: D1, D2, and D3. The HAC dataset comprises seven actions ('sleeping', 'watching tv', 'eating', 'drinking', 'swimming', 'running', and 'opening door') executed by humans (H), animals (A), and cartoon figures (C), resulting in three separate domains: H, A, and C. The HAC dataset includes 3381 video clips sourced from the internet, with approximately 1000 samples per domain. Both datasets offer three modalities: video, audio, and optical flow.

**Baselines.** In our experiments, we compare our CMRF with seven different baselines that can be divided into four groups: 1) Base, naive multi-modal joint training without any domain generalization strategies, 2) SAM [18] and SAGM [25], searching for flat minima in parameter loss landscapes, 3) SWAD [14] and EoA [26], ensemble-based methods for flat minima, and 4) RNA-Net [27] and SimMMDG [28], domain generalization methods specifically designed for MMDG. SAM, SAGM, SWAD and EoA are initially designed for uni-modal DG and we extent them into MMDG. For all methods, we follow [41] and select the model with best validation (in-domain) accuracy to evaluate generalization on test (out-of-domain) data. We report the Top-1 accuracy for all results.

**Implementation Details.** In our framework, we conduct experiments across three modalities: video, audio, and optical flow, adhering to the implementation described in [28]. We leverage the MMAction2 toolkit [44] for our experimental setup. To encode visual information, we utilize the SlowFast network [45], initialized with pre-trained weights on Kinetics-400 [46]. For the audio encoder, we employ ResNet-18 [47], initialized with weights from the VGGSound pre-trained checkpoint [48]. The optical flow encoder uses the SlowFast network's slow-only pathway with Kinetics-400 pre-trained weights. The dimensions of the uni-modal feature $h$ are 2304 for video, 512 for audio, and 2048 for optical flow. For the projector $Proj_k(\cdot)$, we implement a multi-layer perceptron with two hidden layers of size 2048 and output size 128. We use the Adam optimizer [49] with a learning rate of 0.0001 and a batch size of 16. The scalar temperature parameter $\tau$ is set to 0.1. Additionally, we set $\lambda_1 = 2.0$, $\lambda_2 = \lambda_3 = 3.0$, $\alpha$ in the Beta distribution to 0.1, and the SMA start iteration $t_0$ to 400 for EPIC-Kitchens and 100 for HAC respectively. All experiments were conducted on an NVIDIA GeForce RTX 3090 GPU with a 3.9-GHz Intel Core i9-12900K CPU. The model is trained with 15 epochs, taking two hours.

# B    More Results

**Uni-modal in-domain validation performance.** Modal competition refers to the mutual inhibition between modalities in joint training, which is reflected in in-domain performance straightforwardly as studied in previous literature. In Tab.8 we give the uni-modal validation results (in-domain) on EPIC-kitchens with video and audio data. Modal competition is manifested in that each single modality of Base performs worse than uni-modal training, which further leads to worse out-of-domain performance as shown in Tab. 9. Our method achieves the best uni-modal in-domain performance, indicating that it optimizes modal competition effectively, which in turn improves the generalization ability to other domains as in Tab. 4.

Table 8: Uni-modal validation (in-domain) performance under multi-modal multi-source DG on EPIC-Kitchens dataset with video and audio data.

|  | Video | | | | Audio | | | |
|---|---|---|---|---|---|---|---|---|
|  | D2, D3 → D1 | D1, D3 → D2 | D1, D2 → D3 | $Avg$ | D2, D3 → D1 | D1, D3 → D2 | D1, D2 → D3 | $Avg$ |
| Uni-modal | 79.58 | 75.58 | 75.19 | 76.78 | **60.32** | 54.29 | 53.16 | 55.92 |
| Base | 75.78 | 73.60 | 72.40 | 73.93 | 54.58 | 52.23 | 49.11 | 51.97 |
| SAM | 77.03 | 73.81 | 73.75 | 74.86 | 54.90 | 51.60 | 49.67 | 52.06 |
| EoA | 78.94 | 73.20 | 75.12 | 75.75 | 56.85 | 52.76 | 52.45 | 54.02 |
| SimMMDG | 80.86 | 74.81 | 74.57 | 76.75 | 54.58 | 53.34 | 52.90 | 53.60 |
| CMRF(ours) | **81.26** | **77.21** | **75.69** | **78.05** | 58.77 | **54.89** | **54.38** | **56.01** |

**Flatness visualization.** To evaluate the loss flatness, we can apply low-frequency perturbation from the Gaussian Distribution on representations, where the variance controls the perturbation strength.

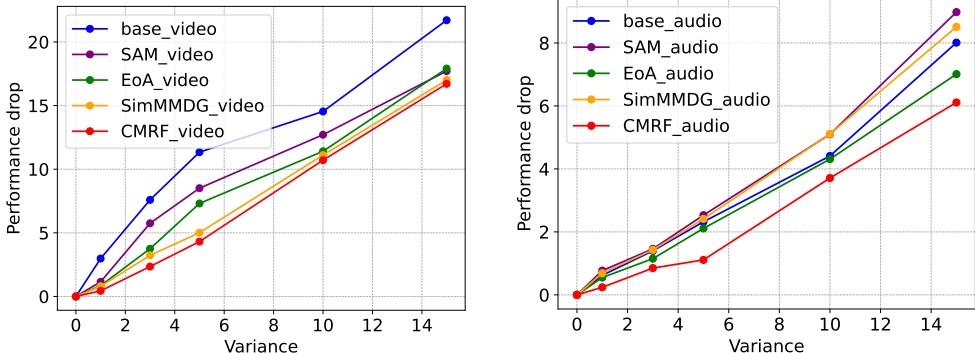

Figure 4: Representation space loss flatness evaluation. We apply gaussian noise to the extracted representations to be the domain shifts. The perturbation variance measures the distance between the perturbed representation and the original representation. We use the performance drop against perturbation variance to measure the sharpness of the landscapes around the minimum, where a larger drop indicates a sharp minimum. The experiments are on EPIC-Kitchens with D2, D3 → D1 of video-audio modalities. Left is the performance drop of video while right is the result of audio.

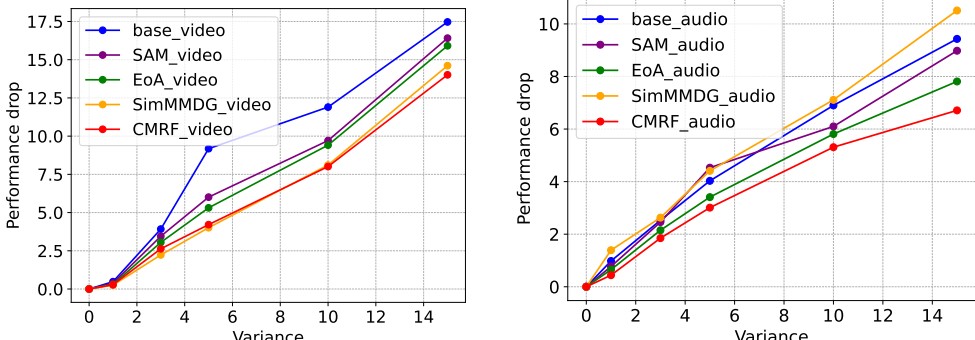

Figure 5: Representation space loss flatness evaluation. EPIC-Kitchens with D2, D3 → D1 of flow-audio modalities. Left is the performance drop of flow while right is the result of audio.

The magnitude of the performance drop indicates how flat the loss is. The results are shown Figs. 4 and 5 below. With the increase of Variance, our method has the smallest performance drop on each modality, indicating that our method achieves flatter loss landscape for both modalities simultaneously and in turn provides flatter multi-modal loss landscape.

**Uni-modal out-of-domain performance.** Here, we give the detailed results of uni-modal performance comparison on EPIC-Kitchens in Tabs. 9, 10, and 11, which form the results in Tab. 4 in the main paper. The results for HAC dataset are demonstrated in Tabs. 12, 13, and 14. Our method can achieve the best uni-modal, as well as multi-modal, performance on both datasets with various modality combinations.

Table 9: Uni-modal performance under multi-modal multi-source DG on EPIC-Kitchens dataset with video and audio data.

| | EPIC-Kitchens | | | | | | | |
|---|---|---|---|---|---|---|---|---|
| | Video | | | | Audio | | | |
| | D2, D3 → D1 | D1, D3 → D2 | D1, D2 → D3 | Avg | D2, D3 → D1 | D1, D3 → D2 | D1, D2 → D3 | Avg |
| Uni-video | 54.02 | **65.60** | 56.57 | 58.73 | - | - | - | - |
| Uni-audio | - | - | - | - | 37.01 | 40.40 | 42.71 | 40.04 |
| Base | 53.33 | 62.00 | 54.62 | 56.65 | 36.32 | 34.60 | 44.95 | 38.62 |
| SAM [18] | 55.86 | 61.20 | 59.34 | 58.80 | 33.32 | 35.87 | 44.13 | 37.77 |
| EoA [26] | 53.82 | 63.14 | 55.67 | 57.54 | **38.16** | 37.04 | 43.55 | 39.70 |
| SimMMDG [28] | 54.67 | 63.75 | 59.87 | 59.43 | 32.21 | 34.98 | 48.12 | 38.43 |
| CMRF (ours) | **56.79** | 64.10 | **61.09** | **60.66** | 37.94 | **43.32** | 48.12 | **43.13** |

Table 10: Uni-modal performance under multi-modal multi-source DG on EPIC-Kitchens dataset with video and optical flow data.

| | EPIC-Kitchens | | | | | | | |
|---|---|---|---|---|---|---|---|---|
| | Video | | | | Flow | | | |
| | D2, D3 → D1 | D1, D3 → D2 | D1, D2 → D3 | Avg | D2, D3 → D1 | D1, D3 → D2 | D1, D2 → D3 | Avg |
| Uni-video | 54.02 | **65.60** | 56.57 | 58.73 | - | - | - | - |
| Uni-flow | - | - | - | - | **56.55** | 62.00 | 56.36 | 58.30 |
| Base | 47.82 | 61.47 | 56.57 | 55.28 | 52.18 | 60.53 | 54.62 | 55.78 |
| SAM [18] | 54.94 | 63.87 | 60.47 | 59.76 | 52.64 | 59.47 | 56.03 | 56.05 |
| EoA [26] | 51.67 | 63.33 | 57.48 | 57.49 | 53.04 | 62.13 | 56.34 | 57.17 |
| SimMMDG [28] | 50.54 | 60.76 | 59.77 | 57.02 | 50.33 | 62.89 | 53.58 | 55.60 |
| CMRF (ours) | **55.63** | 62.13 | **61.74** | **59.83** | 53.79 | **63.10** | **58.11** | **58.33** |

Table 11: Uni-modal performance under multi-modal multi-source DG on EPIC-Kitchens dataset with optical flow and audio data.

| | EPIC-Kitchens | | | | | | | |
|---|---|---|---|---|---|---|---|---|
| | Flow | | | | Audio | | | |
| | D2, D3 → D1 | D1, D3 → D2 | D1, D2 → D3 | Avg | D2, D3 → D1 | D1, D3 → D2 | D1, D2 → D3 | Avg |
| Uni-flow | **56.55** | 62.00 | 56.36 | 58.30 | - | - | - | - |
| Uni-audio | - | - | - | - | 37.01 | 40.40 | 42.71 | 40.04 |
| Base | 51.72 | 57.73 | 55.13 | 54.86 | 36.32 | 38.00 | 43.94 | 39.42 |
| SAM [18] | 53.56 | 60.00 | 56.90 | 56.82 | 37.70 | 38.93 | 44.43 | 40.35 |
| EoA [26] | 54.43 | 59.87 | 57.67 | 57.32 | 38.16 | 40.40 | 41.85 | 40.14 |
| SimMMDG [28] | 56.27 | 61.58 | 56.79 | 58.21 | 35.82 | 36.49 | 47.78 | 40.03 |
| CMRF (ours) | 56.27 | **63.37** | **59.24** | **59.63** | **40.00** | **41.47** | **49.28** | **43.58** |

Table 12: Uni-modal performance under multi-modal multi-source DG on HAC dataset with video and audio data.

| | HAC | | | | | | | |
|---|---|---|---|---|---|---|---|---|
| | Video | | | | Audio | | | |
| | A, C → H | H, C → A | H, A → C | Avg | A, C → H | H, C → A | H, A → C | Avg |
| Uni-video | 73.29 | 77.11 | 53.80 | 68.07 | - | - | - | - |
| Uni-audio | - | - | - | - | 28.26 | 38.09 | **32.11** | 32.81 |
| Base | 72.83 | 72.72 | **57.26** | 67.60 | **31.16** | 36.50 | 26.06 | 31.24 |
| SAM [18] | 71.84 | 78.41 | 55.13 | 68.46 | 30.25 | 39.20 | 25.23 | 31.56 |
| CMRF (ours) | **74.64** | **83.52** | 53.46 | **70.54** | 30.43 | **44.32** | 29.82 | **34.86** |

Table 13: Uni-modal performance under multi-modal multi-source DG on HAC dataset with video and optical flow data.

| | HAC | | | | | | | |
| | Video | | | | Flow | | | |
| | A, C → H | H, C → A | H, A → C | *Avg* | A, C → H | H, C → A | H, A → C | *Avg* |
|---|---|---|---|---|---|---|---|---|
| Uni-video | 73.29 | 77.11 | **53.80** | 68.07 | - | - | - | - |
| Uni-flow | - | - | - | - | 57.97 | 58.52 | **43.12** | 53.20 |
| Base | 72.10 | 74.43 | 46.33 | 64.29 | 56.16 | 53.98 | 35.78 | 48.64 |
| SAM [18] | 74.64 | 78.98 | 49.08 | 67.57 | 53.62 | 50.00 | 37.15 | 46.92 |
| CMRF (ours) | **77.90** | **79.84** | 48.33 | **68.69** | **63.04** | **62.50** | 37.78 | **54.44** |

Table 14: Uni-modal performance under multi-modal multi-source DG on HAC dataset with optical flow and audio data.

| | HAC | | | | | | | |
| | Flow | | | | Audio | | | |
| | A, C → H | H, C → A | H, A → C | *Avg* | A, C → H | H, C → A | H, A → C | *Avg* |
|---|---|---|---|---|---|---|---|---|
| Uni-flow | 57.97 | **58.52** | 43.12 | 53.20 | - | - | - | - |
| Uni-audio | - | - | - | - | 28.26 | 38.07 | 32.11 | 32.81 |
| Base | 55.86 | 56.82 | 41.50 | 51.39 | 27.35 | 37.34 | 26.15 | 30.28 |
| SAM | 60.51 | 55.13 | **48.62** | 54.75 | 29.16 | 40.04 | 30.23 | 32.14 |
| CMRF (ours) | **61.59** | 57.95 | 47.49 | **55.68** | **31.88** | **41.48** | **33.03** | **35.46** |

**Validation and test comparison with uni-modal training.** In Tab. 15 and Tab. 16, we report the in-domain validation and out-of-domain test results on EPIC-kitchens and HAC datasets for each modality. We can see that for each modality, its validation performance is strongly positive correlated to its test performance, i.e., modalities that perform better on the validation set usually perform better on the test set. This provides empirical support for us to use validation set accuracy in Eq. 11 to evaluate the generalization ability of different modalities.

Table 15: Uni-modal validation performance vs. test performance on EPIC-Kitchens dataset.

| | Validation | | | | Test | | | |
| | D2, D3 → D1 | D1, D3 → D2 | D1, D2 → D3 | *Avg* | D2, D3 → D1 | D1, D3 → D2 | D1, D2 → D3 | *Avg* |
|---|---|---|---|---|---|---|---|---|
| Video | 79.58 | 75.58 | 75.19 | 76.78 | 54.02 | 65.60 | 56.57 | 58.73 |
| Flow | 74.94 | 72.04 | 72.57 | 73.18 | 56.55 | 62.00 | 56.36 | 58.30 |
| Audio | 60.32 | 54.29 | 53.16 | 55.92 | 37.01 | 40.40 | 42.71 | 40.04 |

Table 16: Uni-modal validation performance vs. test performance on HAC dataset.

| | Validation | | | | Test | | | |
| | A, C → H | H, C → A | H, A → C | *Avg* | A, C → H | H, C → A | H, A → C | *Avg* |
|---|---|---|---|---|---|---|---|---|
| Video | 90.10 | 88.66 | 93.58 | 90.78 | 73.29 | 77.11 | 53.80 | 68.07 |
| Flow | 74.11 | 72.87 | 80.53 | 78.54 | 57.97 | 58.52 | 43.12 | 53.20 |
| Audio | 56.09 | 49.19 | 55.09 | 53.46 | 28.26 | 38.07 | 32.11 | 32.81 |

