# OpenReview forum: "Cross-modal Representation Flattening for Multi-modal Domain Generalization"
_NeurIPS.cc/2024/Conference — NeurIPS 2024 poster_

### Official Review · Reviewer_8XoV · 2024-07-09

**Soundness:** 3
**Presentation:** 3
**Contribution:** 2
**Rating:** 6
**Confidence:** 4

**Summary:**

This paper extends the analysis of unimodal flatness to the domain of multi-modal domain generalization (MMDG) for the first time, and proposes the Cross-Modal Representation Flattening (CMRF) method. By constructing a shared representation space and cross-modal knowledge distillation, it addresses the issues of competition and inconsistency in flatness across multiple modalities. Experimental results demonstrate that CMRF significantly outperforms all baseline methods in the multi-modal multi-source domain generalization setting, effectively enhancing the model's generalization capability. In particular, it achieves an average improvement of up to 3.52% in the video-audio modality combination.

**Strengths:**

1. The new Cross-Modal Representation Flattening (CMRF) method is used for multi-modal domain generalization. This method constructs a consistent flat loss region in the representation space and utilizes cross-modal knowledge transfer to enhance the generalization ability of each modality, which is a novel approach in the multi-modal domain.

2. The paper delves into two key issues in multi-modal domain generalization: inter-modal competition and inconsistent flatness between modalities. This analysis lays a solid foundation for proposing effective solutions.

3. The paper mentions that applying SAM can only improve the generalization of the better modalities in multi-modal networks, while it has little to no benefit and may even be harmful for the weaker modalities. This indicates that traditional SAM-based single-modal methods, due to generalization gaps, fail to find the flat minimum value where each modality coexists, leading to inconsistent flatness and thus not fully utilizing the generalization ability of all modalities. The analysis and conclusions are correct.

**Weaknesses:**

1. The paper mentions using simple moving average to build the teacher network, but how to extract knowledge from the mixed representation and how to transfer this knowledge to each modality is not clear enough.

2. The paper points out that inter-modal competition is a key issue, but the specific performance of the competition and the detailed evaluation of the proposed mitigation strategies are not thorough. It is hoped that the authors can provide more quantitative analysis on inter-modal competition and validation of the effectiveness of the mitigation strategies.

**Questions:**

1. In Section 3.3, the authors proposed a method for cross-modal representation flattening, but further explanation is needed on how to construct interpolation representations and how to optimize these interpolations to achieve flattening of the loss region.

2. The authors should provide more details on how to dynamically adjust weights based on the differences in the generalization ability of each modality.

**Limitations:**

Authors have adequately addressed the limitations.

---

> ### Author Rebuttal · Authors · 2024-08-05
>
> We thank reviewer for valuable comments. We respond below to each of the concerns.
>
> >1. The paper mentions using simple moving average to build the teacher network, but how to extract knowledge from the mixed representation and how to transfer this knowledge to each modality is not clear enough.
>
> **Response:** The reason why using the mixed representation promotes multimodal domain generalization is twofold. Firstly, the mixed representations are constructed based on the moving averaged teacher network, which has flatter loss landscape compared with student network [A]. Thus, the representations from teacher network naturally leads to better generalization. Secondly, mixing the representations from one modality with representations from other modality means applying perturbations, and the applied perturbations guide the current modality to learn about other modality. Therefore, by distillation, we extract knowledge from the mixed representation and transfer them to each modality. Moreover, as the representations of each modality learn towards these interpolations, we optimize them by uni-modal classification losses, which ensures that when the losses of interpolations are large, we can also optimize them to lower losses, so that the representation loss landscape between modalities can be flattened.
>
> > 2. The paper points out that inter-modal competition is a key issue, but the specific performance of the competition and the detailed evaluation of the proposed mitigation strategies are not thorough. It is hoped that the authors can provide more quantitative analysis on inter-modal competition and validation of the effectiveness of the mitigation strategies.
>
> **Response:**  Modality competition refers to the mutual inhibition between modalities in joint training, which is reflected in in-domain performance straightforwardly as studied in previous literature. Below we give the uni-modal validation results (in-domain) on EPIC-kitchens with video and audio data. Modal competition is manifested in that each single modality of Base performs worse than uni-modal training, which further leads to worse out-of-domain performance for each modality, as in the results of Tabs. 1,4, where the generalization ability of each modality degrades under different modality combinations and different datasets, indicating the existance of modality competition.
> More results of the in-domain validation accuracy will be released later.
>
> |||Video||||Audio|||
> |:--------------:|:----------------:|:----------------:|:----------------:|:------------:|:----------------:|:----------------:|:----------------:|:------------:|
> ||D2,D3->D1|D1,D3->D2|D1,D2->D3| Avg |D2,D3->D1|D1,D3->D2|D1,D2->D3| Avg|
> |Uni-modal|79.58|75.58|75.19|76.78|**60.32**|54.29|53.16|55.92|
> |Base|75.78|73.60|72.40|73.93|54.58|52.23|49.11|51.97|
> |SAM|77.03|73.81|73.75|74.86|54.90|51.60|49.67|52.06|
> |EoA |78.94|73.20|75.12|75.75|56.85|52.76|52.45|54.02|
> |SimMMDG|80.86|74.81|74.57|76.75|54.58|53.34|52.90|53.60|
> |CMRF(ours)|**81.26**|**77.21**|**75.69**|**78.05**|58.77|**54.89**|**54.38**|**56.01**|
>
> In contrast, our method achieves the best uni-modal in-domain performance as shown above, indicating that **it optimizes modality competition effectively**, which in turn improves the generalization ability to other domains as in Tab. 4.  The detailed results with each domain as target and more results on HAC dataset are shown in Tab. 8-13 in appendix, demonstrating modality competition and our approach really mitigates this problem comprehensively.
>
> > 3. In Section 3.3, the authors proposed a method for cross-modal representation flattening, but further explanation is needed on how to construct interpolation representations and how to optimize these interpolations to achieve flattening of the loss region.
>
> **Response:** How to construct interpolation representations can be explained by Eq. 8. The extracted representations from teacher network will sum weighted, where the weight $\delta$ is sampled from distribution Beta($\alpha$, $\alpha$). Here we set $\alpha <1$, so that $\delta$ is biased towards 0 or 1. Considering the semantic gap between modalities, we let interpolation closer to student modality act as its teacher signal.  For example, for two representations $z_M$ and $z_N$ from modality M and N, one interpolation is $z_{M,N}=\delta z_M + (1 - \delta ) z_N $, if $\delta = 0.8$ (close to M), $z_{M,N}$ will be the teacher signal for modality M and if $\delta = 0.2$ (close to N), $z_{M,N}$ will be the teacher signal for modality N, the same as Eq. 9.
>
> How to optimize these interpolations depends on two losses: the distillation loss and the uni-modal classification loss (L186, L187). As our response for Question 1, by distillation, we make the representations of each modality learn towards interpolations. Then by uni-modal classification losses, if the losses of interpolations are large, we can optimize them to lower losses, so that the representation loss landscape between modalities can be flattened.
>
> > 4. The authors should provide more details on how to dynamically adjust weights based on the differences in the generalization ability of each modality.
>
> **Response:** The weight is adjusted under different epochs. We use the teacher network with its uni-modal classifier to get each uni-modal average validation accuracy on validation set, i.e. $\hat{A}_k$. And if the performance gap between two modalities is greater than a certain threshold $\mu$, we put a larger weight (1.0) on the better modality to make it play a greater role in the distillation process and put a smaller weight (0.5) for the other.
>
> [A] Devansh Arpit,et al.Ensemble of averages: Improving model selection and boosting performance in domain generalization. NIPS 2022.

---

> ### Comment · Reviewer_8XoV · 2024-08-13
>
> The authors' rebuttal dispelled some of my concerns, and I choose to maintain the score.

---

### Official Review · Reviewer_iu3y · 2024-07-11

**Soundness:** 3
**Presentation:** 3
**Contribution:** 3
**Rating:** 6
**Confidence:** 5

**Summary:**

The paper addresses the problem of multi-modal domain generalization (MMDG). The authors identify two main challenges in MMDG: modality competition and discrepant uni-modal flatness and propose a novel method called Cross-Modal Representation Flattening (CMRF). This method optimizes the representation-space loss landscapes instead of the traditional parameter space, allowing for direct connections between modalities. The authors demonstrate the effectiveness of CMRF through extensive experiments on benchmark datasets EPIC-Kitchens and Human-Animal-Cartoon (HAC), showing superior performance in both multi-source and single-source settings.

**Strengths:**

1. The introduction of Cross-Modal Representation Flattening is novel to address the challenges of MMDG
2. The paper provides a detailed analysis of modality competition and discrepant uni-modal flatness, which are crucial factors in multi-modal generalization
3. Comprehensive evaluation demonstrates the effectiveness of the proposed methods
4. The paper is very well written and easy to follow

**Weaknesses:**

1. Are there any visualization or quantitative results to show that after CMRF training the loss landscape of different modalities get consistent flat region?
2. What is the influence of α in the Beta distribution? What if it is larger than 1?

**Questions:**

Are there any visualization or quantitative results to show that after CMRF the loss landscape of different modalities get consistent flat region? What is the influence of α in the Beta distribution? What if it is larger than 1?

**Limitations:**

Currently, the paper need to test on validation set to estimate generalization of each modality for Eq. 11, which can be time-consuming with the scale increase of validation set.

---

> ### Author Rebuttal · Authors · 2024-08-05
>
> We thank reviewer for valuable comments. We respond below to each of the concerns.
>
> > 1. Are there any visualization or quantitative results to show that after CMRF training the loss landscape of different modalities get consistent flat region?
>
> **Response**: Thanks a lot for the valuable comments. We can evaluate the loss flatness by applying low-frequency perturbation from the Gaussian Distribution on representations as in [A], where the Variance controls the perturbation strength. The magnitude of the performance drop indicates how flat the loss is. The results are shown Figs.1,2 in rebuttal.pdf. With the increase of Variance, the performance drop of different modalities grows quickly. Although previous SAM can obtain flatter loss landscape for the better modality (video or optical flow), its improvement to weak modality (audio) is low or even harmful. The same phenomenon also exists in some multimodal domain generalization method such as SimMMDG. In contrast, our method has the smallest performance drop on each modality, indicating that our method achieves flatter loss landscape for both modalities simultaneously, i.e., getting consistent flat region. More results on different modality combinations and different datasets will be released later.
>
> > 2. What is the influence of α in the Beta distribution? What if it is larger than 1?
>
> **Response**: In this paper, we let the parameters in Beta($\alpha$, $\beta$) equal, i.e., $\alpha=\beta$, so it becomes Beta($\alpha$, $\alpha$). In this way,  the distribution is symmetric around the midpoint of the interval [0, 1]. When $\alpha <1$, the Beta distribution is U-shaped, with more weight towards the boundaries (0 and 1). When $\alpha >1$, the Beta distribution becomes symmetric and bell-shaped around 0.5. The larger the value of $\alpha$, the more peaked the distribution becomes around 0.5. In this paper, we want to flatten each modality's loss landscape by distilling knowledge from the mixed multimodal representations. However, because of the knowledge heterogeneity between modalities, if the mixed representations contains much more inconsistent knowledge from other modalities, the cross-modal distillation will lead to worse performance [B]. Therefore, we want the mixed teacher information used for distillation to contain more information about the student modality, i.e., the values sampled from the beta distribution should be biased towards 0 or 1. And if $\alpha >1$, the values sampled from the beta distribution are more likely to be 0.5, where the knowledge of two modalities is mixed equally. In this case, if the other modality contains knowledge that is inconsistent with the student modality, the performance of distillation will be degraded. This is also the example represented by Fixed Mix (mixing ratio is 0.5) in Tab. 6, which is worse than our method with $\alpha=0.1$.
>
> [A] Yixiong Zou, et al. Flatten Long-Range Loss Landscapes for Cross-Domain Few-Shot Learning. CVPR 2024.
>
> [B] Zihui Xue, et al.The Modality Focusing Hypothesis: Towards Understanding Crossmodal Knowledge Distillation. ICLR 2023.

---

> > ### Comment · Reviewer_iu3y · 2024-08-11
> >
> > Thank the authors for their rebuttal. Most of my concerns have been well-addressed and I thus increased my score to 6.

---

### Official Review · Reviewer_eDBu · 2024-07-12

**Soundness:** 2
**Presentation:** 3
**Contribution:** 2
**Rating:** 5
**Confidence:** 3

**Summary:**

In this paper, the authors identify two primary limitations in multi-modal domain generalization (MMDG): modality competition and discrepant uni-modal flatness. To address these challenges, they propose a novel approach called Cross-Modal Representation Flattening (CMRF). CMRF constructs interpolations by mixing multi-modal representations from a teacher model and uses feature distillation to optimize high-loss regions between modalities. It adopts a supervised multi-modal contrastive loss to enhance the flat loss region in the shared feature space. Furthermore, CMRF employs adaptive weighting based on modal validation set accuracy to better utilize each modality. The effectiveness of the proposed method is validated through extensive experiments.

**Strengths:**

1. Flatness analysis within multi-modal domain generalization (MMDG) is innovative. Identifying modality competition and discrepant uni-modal flatness as key limitations offers a fresh perspective on the challenges in the field and has the potential to advance understanding in the broader area of multi-modal learning.
2. Extensive experiments in various modality combinations and settings demonstrate the effectiveness of the proposed model.

**Weaknesses:**

1. This paper lacks a detailed discussion on the practical implementation aspects of the proposed method, such as computational requirements and scalability.
2. Although the authors' claims about addressing modality competition and discrepant uni-modal flatness are intuitive, the paper does not adequately address how to evaluate the loss flatness for multi-modal data. Additionally, the proposed method lacks sufficient evidence and empirical results to convincingly demonstrate its effectiveness in optimizing these specific limitations within multi-modal domain generalization.

**Questions:**

1. What does the comparative data in Table 1 represent?

**Limitations:**

The authors have discussed the limitations in the paper.

---

> ### Author Rebuttal · Authors · 2024-08-05
>
> We thank reviewer for valuable comments. We respond below to each of the concerns.
>
> > 1. This paper lacks a detailed discussion on the practical implementation aspects of the proposed method, such as computational requirements and scalability.
>
> **Response:**
>
> **1) Computational requirements:** Firstly, the Projector (3-layer MLP) for each modality is very small in our method and is also used in previous method such as SimMMDG, indicating that additional computation from more parameters is negligible. Secondly, although our method requires more computation from teacher model, it only requires the forward propagation but not gradient backpropagation on it, so the additional computation is also limited.
> Finally, the main additional computational costs come from testing on validation set for adaptive weights Eq.11, as discussed in our Limitation section. The one epoch training time on D2,D3->D1 with video-audio data from EPIC-kitchens of different methods are shown below (in seconds). SWAD and our CMRF need more computation as they both require testing on validation set (The value in parentheses indicates the validation time). Searching for an efficient method to evaluate the generalization of each modality would be our future work, such as add low-frequency noise to stimulate domain shift for evaluation.
> | Base | SAM | SAGM | SWAD | EoA | RNA-Net | SimMMDG | CMRF (ours) |
> |--------|---------|----------|-----------|-------|--------------|----------------|-------------------|
> | 308.5 | 346.4 | 345.6 | 367.9 | 309.2 | 311.3 | 327.7 | 369.9 (59.4) |
>
>
> **2) Scalability:** Our method has no restrictions on the type and number of modalities. Since our method focuses on representation space other than parameter space, the condition that our method can be applied is that a consistent representation space of multiple modalities is available, which is definitely satisfied in multimodal deep learning by mapping with neural networks. This also demonstrates the flexibility of our approach to different model structures. In this paper, our experiments include the three common modalities (video, audio, optical flow), as well as different numbers of modalities, to verify our claims. All these show that our method has good scalability.
>
> > 2. Although the authors' claims about addressing modality competition and discrepant uni-modal flatness are intuitive, the paper does not adequately address how to evaluate the loss flatness for multi-modal data. Additionally, the proposed method lacks sufficient evidence and empirical results to convincingly demonstrate its effectiveness in optimizing these specific limitations within multi-modal domain generalization.
>
> **Response:** Thanks a lot for the valuable comments. We can evaluate the loss flatness by applying low-frequency perturbation from the Gaussian Distribution on representations as in [A], where the Variance controls the perturbation strength. The magnitude of the performance drop indicates how flat the loss is. The results are shown Figs.1,2 in rebuttal.pdf. With the increase of Variance, our method has the smallest performance drop on each modality, indicating that our method achieves flatter loss landscape for both modalities simultaneously and in turn provides flatter multi-modal loss landscape.
>
> In order to better demonstrate the effectiveness of our approach in solving the two specific limitations, please allow us to restate the two problems. **Modal competition** refers to the mutual inhibition between modalities in joint training, which is reflected in in-domain performance straightforwardly as studied in previous literature. Below we give the uni-modal validation results (in-domain) on EPIC-kitchens with video and audio data. Modal competition is manifested in that each single modality of Base performs worse than uni-modal training, which further leads to worse out-of-domain performance as shown in Tab. 8 in appendix. Our method achieves the best uni-modal in-domain performance, indicating that **it optimizes modal competition effectively**, which in turn improves the generalization ability to other domains as in Tab. 4.
>
> |||Video||||Audio|||
> |:--------------:|:----------------:|:----------------:|:----------------:|:------------:|:----------------:|:----------------:|:----------------:|:------------:|
> ||D2,D3->D1|D1,D3->D2|D1,D2->D3| Avg |D2,D3->D1|D1,D3->D2|D1,D2->D3| Avg|
> |Uni-modal|79.58|75.58|75.19|76.78|**60.32**|54.29|53.16|55.92|
> |Base|75.78|73.60|72.40|73.93|54.58|52.23|49.11|51.97|
> |SAM|77.03|73.81|73.75|74.86|54.90|51.60|49.67|52.06|
> |EoA |78.94|73.20|75.12|75.75|56.85|52.76|52.45|54.02|
> |SimMMDG|80.86|74.81|74.57|76.75|54.58|53.34|52.90|53.60|
> |CMRF(ours)|**81.26**|**77.21**|**75.69**|**78.05**|58.77|**54.89**|**54.38**|**56.01**|
>
> **Discrepant uni-modal flatness** refers to the fact that the generalization gaps between modalities make it difficult to optimize flat regions for each modality simultaneously. According to our analysis in Sec. 3.2 and results in Tabs. 1, 4 and above tabel, previous SAM-based methods contribute more to the better modality and contribute less or even are harmful to weak modalities.  The Fig. 1 in rebuttal.pdf also show that SAM obtains flatter loss for better modality (video) and sharper loss for weak modality (audio). In contrast, our method **achieves flatter loss landscape simultaneously for each modality, leading to consistent flatness**.
>
> The above shows that the effectiveness of our method for both modal competition and Discrepant uni-modal flatness, and the improvement of multimodal domain generalization comes from solving both problems at the same time, not just one of them.
>
> > 3. What does the comparative data in Table 1 represent?
>
> Data in Tab. 1 are the average results by treating each domain as target domain, e.g., each data is the average of results from A,C→H H,C→A H,A→C on HAC.
>
> [A] Yixiong Zou, et al. Flatten Long-Range Loss Landscapes for Cross-Domain Few-Shot Learning. CVPR 2024.

---

> > ### Comment · Reviewer_eDBu · 2024-08-13
> >
> > Thank the authors for addressing my concerns. I tend to increase my score to 5.

---

### Author Rebuttal · Authors · 2024-08-05

We thank all reviewers for their positive feedback:

1. A fresh perspective of modality competition and discrepant uni-modal flatness for MMDG [eDBu, iu3y, 8XoV];
2. Flatness analysis within multi-modal domain generalization (MMDG) and the proposed method for flatting representation loss landscape are innovative [eDBu, 8XoV];
3. Good writing and easy to follow [iu3y]
4. Extensive experiments demonstrate the effectiveness of the proposed model [eDBu, iu3y].

We respond to each reviewer's comments below. We provide some figures in the rebuttal.pdf.

---

### Decision · Program_Chairs · 2024-09-25

**Decision:**

Accept (poster)

**Comment:**

The paper identifies two main issues related to multi-modal domain generalization (MMDG) methods -- modality competition and discrepant uni-modal flatness. The authors propose a CMRF method, which mixes multi-modal representations from a teacher model and uses feature distillation to optimize high-loss regions between modalities. It uses a supervised multi-modal contrastive loss to enhance the flat loss region in the shared feature space. Furthermore, CMRF does adaptive weighting based on modal validation set accuracy to better utilize each modality. The effectiveness of the proposed method is validated through experiments.

The reviews of the paper are overall positive. The paper is recommended for acceptance as a poster paper. The authors are advised to address the remaining feedback of the reviewers, which were not handled during author feedback time.